# GROOT: CORRECTIVE REWARD OPTIMIZATION FOR GENERATIVE SEQUENTIAL LABELING

## ABSTRACT

Sequential labeling is a fundamental NLP task, forming the backbone of many applications. Supervised learning of Seq2Seq models has shown great success on these problems. However, the training objectives are still significantly disconnected with the metrics and desiderata we care about in practice. For example, a practical sequence tagging application may want to optimize for a certain precision-recall trade-off (of the *top-k predictions*) which is quite different from the standard objective of maximizing the likelihood of the *gold labeled sequence*. Thus to bridge this gap, we propose **GROOT** – a simple yet effective framework for **G**enerative **R**eward **O**ptimization **O**f **T**ext sequences. GROOT works by training a generative sequential labeling model to match the decoder output distribution with that of the (black-box) reward function. Using an iterative training regime, we first *generate* prediction candidates, then *correct* errors in them, and finally *contrast* those candidates (based on their reward values). As demonstrated via extensive experiments on four public benchmarks, GROOT significantly improves all reward metrics. Furthermore, GROOT leads to improvements of the overall decoder distribution as evidenced by the quality gains of the top-$k$ candidates.

## 1 INTRODUCTION

Sequential labeling tasks are ubiquitous among NLP applications. Tasks ranging from syntactic analysis (e.g., POS tagging and phrase chunking) to semantic analysis (e.g., named entity recognition, slot filling, and query segmentation), are critical components in end-to-end applications, such as search engines and goal-oriented dialog systems.

Advances in pretraining of generative language models (LMs) like T5 (Raffel et al., 2020) and mT5 (Xue et al., 2021) have enabled us to use the same training

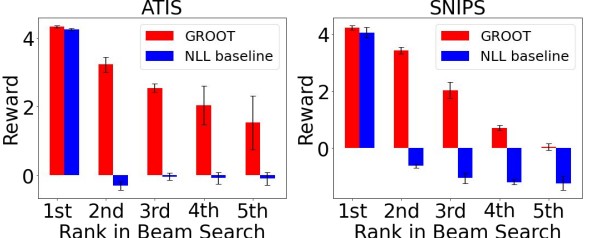

Figure 1: Results for our model (GROOT) vs the NLL baseline demonstrating the precipitous drop-off in quality of NLL model predictions outside the top-1.

strategy seamlessly across these diverse sequence labeling tasks. We can fine-tune a pretrained LM by maximizing the likelihood of generating the ground-truth (human annotated) labeled data.

However, in practice, the metrics and constraints we may care about remain fairly disconnected from the standard Negative Log-Likelihood (**NLL**) objective used to train these models. To understand this better, consider an example of an entity recognition model within an e-commerce system. This model would be typically trained on data of the following form:



**Input**: black & decker blender under 100
**Label**: [BRAND black & decker] [PRODUCT blender] [PRICE under 100]



While this e-commerce pipeline could utilize the model's predictions in different ways, a likely use is in retrieving candidates that match the predicted annotations. However, with models being imperfect, even well-trained models may make errors like:

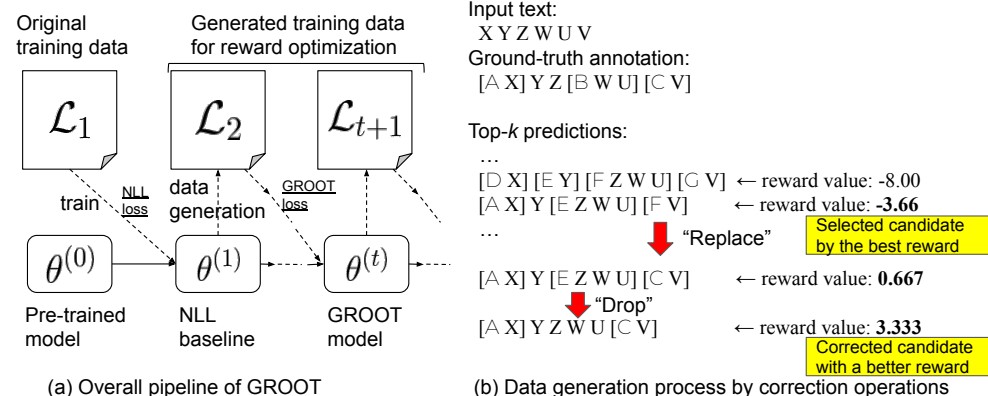

Figure 2: (a) An overview of the GROOT pipeline and (b) training data generation process.

| NLL | (a) Model predictions for an example from the SNIPS *validation* set |
|---|---|
| 0.213 | book [party_size_number seven] in [spatial_relation neighboring] **[geographic_poi moorpark]** |
| 0.564 | book [party_size_number seven] in [spatial_relation neighboring] **[poi moorpark]** |
| 1.311 | book [party_size_number seven] in [spatial_relation neighboring] [city moorpark] (*) |
| 1.443 | book [party_size_number seven] in [spatial_relation neighboring] **[object_name moorpark]** |
| 1.878 | book [party_size_number seven] in **[spatial_relation neighboring moorpark]** |

| NLL | (b) Model predictions for an example from the SNIPS *training* set |
|---|---|
| 0.0003 | add [artist stephen mcnally] to [playlist confidence boost] (*) |
| 3.9628 | add **[entity_name stephen mcnally]** to [playlist confidence boost] |
| 4.0684 | add [artist stephen mcnally] to **[playlist confidence]** boost |
| 4.6491 | add [artist stephen mcnally] to **[entity_name confidence boost]** |
| 5.3953 | add [artist stephen mcnally] to **[album confidence boost]** |

Table 1: Top-5 predictions of NLL baseline model (with errors **bolded**, and perfect prediction marked with (*)).

**Incorrect prediction**: [COLOR black] & decker [PRODUCT blender] [PRICE under 100]

Thus such a retrieval usecase may require a desired precision-recall balance, since *precision* errors (*i.e.,* incorrectly annotated spans) could lead to catastrophic failures downstream – perhaps incorrectly filtering only "black" colored blenders in the above example. Unfortunately, current models do not allow us to optimize for or incorporate such complex metrics or trade-offs.

Such errors are in fact commonplace as seen in Table 1 and empirically in our results. The issue is further exacerbated when we go beyond the top-1 prediction as seen in Figure 1 – with a drastic drop-off in the quality of predictions outside the top-1.

In addition to identifying this shortcoming of NLL-based models, we make the following contributions in this paper:

- We propose **GROOT** – a simple yet effective framework for training sequence labeling models to optimize (black-box) reward metrics. GROOT takes a generative sequential labeling model to learn the reward metrics associated with the output space of sequences.
- We propose CML – a new *Corrective Margin Loss* function – that contrasts candidates with differing reward metrics, to enable the model to better understand the reward space.
- We show that simply relying on candidates sampled from the decoder output distribution – as proposed in prior work for machine translation (Shu et al., 2021) – does not work (and often worsens reward scores significantly).
- To enable principled and targeted *exploration* of the output reward space, we introduce a *correction* function-based approach – *correcting* errors in predictions to explore the space.
- Extensive experiments over 4 public datasets demonstrate that GROOT significantly improves rewards over competitive baselines.
- Furthermore, we demonstrate that GROOT learns a better overall decoder distribution with significant gains in correlation with reward scores.

With extensive experiments aimed at understanding the value of each component of GROOT, we believe our work can help significantly influence practical applications of sequence labeling models.

## 2 RELATED WORK

### 2.1 GENERATIVE SEQUENTIAL LABELING

Sequential labeling typically involves tagging spans of the input with their corresponding label. The emergence of Seq2Seq learning (Sutskever et al., 2014) has seen common NLP tasks like POS tagging, chunking, NER and segmentation cast as Seq2Seq text generation tasks (Vinyals et al., 2015; FitzGerald, 2020; Raffel et al., 2020). More formally, given an input text $x$, we can generate the sequential text label $y$ as $y = \arg\max_{y'} p_\theta(y'|x)$, where the output sequence probability $p_\theta(y'|x)$ is computed by a Seq2Seq model, denoted by the model parameters $\theta$. Arbitrary text formats are acceptable for $x$ and $y$, as long as we can interpret the output. For example, Raman et al. (2022) investigated various formats, and showed that such a generative approach using pre-trained Seq2Seq models outperforms word-level classification models (e.g., mBERT (Devlin et al., 2018)).

To train the Seq2Seq models, we typically use a (human-annotated) ground-truth dataset: a set of training examples, $\mathcal{D}^{\text{train}}$, where each example $e$ is a pair of an input text $x$ and its ground-truth annotation $y^*$: $e = (x,\ y^*)$. The most common supervised learning approach (Williams & Zipser, 1989) would then optimize the NLL loss:

$$\text{NLL}_\theta(e) = -\log p_\theta(y^*|x). \tag{1}$$

While the NLL loss-based training offers many advantages and promising results (Athiwaratkun et al., 2020; Raffel et al., 2020; Yan et al., 2021), it also comes with significant drawbacks. Consider the top-5 predictions of an NLL loss-based model (Table 1) on the SNIPS dataset (Coucke et al., 2018). Aside from the clear overfitting issue (commonly observed in these models (Bishop & Nasrabadi, 2006)), we also find that the top-5 predictions have numerous different errors. While any ML model is bound to make prediction errors in practice, we should recognize that different errors affect downstream applications differently. For example, some applications may prefer recall errors / false negatives (i.e., unannotated spans) over precision errors / false positives (i.e., incorrectly annotated spans) or vice-versa. However the NLL loss-based training does not provide for an easy way to incorporate such desiderata or tradeoffs.

### 2.2 REWARD OPTIMIZATION AND CONTRASTIVE LOSS

Recent attempts have been made to more closely correlate the output distribution of text generation models (esp. for machine translation) with a desired reward metric (Ranzato et al., 2016; Shen et al., 2016; Edunov et al., 2018; Choshen et al., 2020). Most notably, Shu et al. (2021) investigated an efficient and effective method of reward optimization for large neural machine translation models. They have proposed a contrastive loss to compare the best and worst prediction candidates among top-$k$ predictions, while previous related studies investigated different combinations of the candidates to define contrastive losses. However, the underlying assumption is that sampling from the decoder distribution allows for effective exploration (and learning) of the metric space. Unfortunately this assumption does hold true in most scenarios as the output distribution of training examples is highly skewed and often contains low-quality predictions outside the top-1 (see example in Table 1). Hence we need alternative ways to achieve meaningful exploration of the metric space.

## 3 PROPOSED FRAMEWORK: GROOT

As alluded to above, solely optimizing the NLL loss may not provide us sufficient exploration of the output metric space. Given a predefined (blackbox) **reward** function, we want the Seq2Seq model to learn an output distribution that more closely correlates with and maximizes reward values.

### 3.1 OVERVIEW

To tackle this issue, we propose GROOT (visualized in Figure 2). GROOT works by iteratively training and improving the reward of decoder outputs. More specifically, starting from the NLL-based initial model $\theta^{(1)}$, in each iteration $2 \le t \le T$ we

- (Section 3.2) Create a new training set $\mathcal{L}_t$ by *correcting* (top-$k$) predictions (on a subset of the training set) of the previous model $\theta^{(t-1)}$. As we show empirically, leveraging this correction function is key to efficiently explore the output space and significantly outperforms exploration based solely on current predictions (Shu et al., 2021).
- (Section 3.3) Train a new improved model $\theta^{(t)}$ using $\mathcal{L}_t$ by applying our proposed *Corrective Margin Loss* that contrasts and orders different candidates based on their rewards.

Henceforth we will denote the predefined (blackbox) reward as $R$, where $R(y,\ y^*)$ denotes the reward for an example prediction $y$ (*optionally* computed using the gold label $y^*$).

## 3.2 DATA GENERATION

For the $t$-th step of GROOT, we create a new training set $\mathcal{L}_t$ from a (random) subset of the training data $\mathcal{L}'_t$. In particular, for each example $e = (x,\ y^*) \in \mathcal{L}'_t$, we do the following:

**- Get top-$k$ predictions**    Using the previous timestep's model $\theta^{(t-1)}$, we perform inference on $\mathcal{L}'_t$ using a beam search to generate the top-$k$ output candidates for each example $e$.

**- Select *correctable* candidate**    Using the input reward function $R(y,\ y^*)$ we compute a reward value for each of the top-$k$ candidates $y$. Based on the rewards we select $\tilde{y}$, using a fixed criterion (e.g. a non-perfect candidate with highest reward in top-$k$). To maximize learning of the output-reward relation, we only select candidates that can be "*corrected*" – as described below.

Based on prior work (Shu et al., 2021), one may assume that simply comparing the top-$k$ allows for sufficient learning of the output space. However, as we show empirically (in Section 6), simply using the top-$k$ predictions alone is unreliable and does not allow for sufficient exploration of the output space – and at times may even worsen results (Table 3). Thus to help us better explore the output space in a guided manner, we introduce a key additional step:

**- Correct erroneous candidate**    To effectively learn what predictions lead to better reward values, we introduce **correction functions** to rectify erroneous predictions. More specifically, a correction function $C(y,\ y*)$ is designed to return a new prediction $y_+$, such that $r_+ = R(y_+,\ y^*)$ is expected be greater than (or equal to) $r = R(y,\ y^*)$.

In other words, given an imperfect prediction $y$ – and *optionally* the gold label $y*$ – the correction function aims to find a prediction that improves upon or fixes errors in $y$. The goal of the correction function is **not** to find the best possible prediction $y_+$ (which would just be the gold label $y^*$) but rather to expose the model to novel (improved) prediction candidates – potentially quite different from the existing model's output distribution. Since correction functions will be used to improve predictions and explore the output space, they could be either algorithmic or themselves model-based. Note that in some applications with complex reward function, we may not be able to easily devise a single correction function that *always* finds an improved prediction. However in such cases, we can instead combine multiple correction functions. We require that the functions *together* can produce a $y_+$ (s.t., $r_+ \geq r$) with high probability. If we cannot find an improved $y_+$ from any correction functions, then we can choose to drop the example instead.

Once we have this improved $y_+$, each example $e$ is expanded to form a new tuple: $\tilde{e} = (x,\ y^*,\ r^*,\ \tilde{y},\ \tilde{r},\ \tilde{y}_+,\ \tilde{r}_+)$, where $r^* = R(y^*,\ y^*)$, $\tilde{r} = R(\tilde{y},\ y^*)$, and $\tilde{r}_+ = R(\tilde{y}_+,\ y^*)$.

## 3.3 TRAINING WITH CORRECTIVE MARGIN LOSS

To help the model best learn the output space of rewards we train the model by contrasting pairs of $\tilde{y}$ and $\tilde{y}_+$. In particular, for each $\tilde{e} \in \mathcal{L}_t$ we use the following **corrective margin loss**:

$$\text{CML}_\theta(x,\ \tilde{y},\ \tilde{r},\ \tilde{y}_+,\ \tilde{r}_+) = \max(0,\ m - \log p(\tilde{y}_+|x) + \log p(\tilde{y}|x)), \tag{2}$$

where $m$ is a margin computed with a hyperparameter $\alpha$: $m = \alpha(\tilde{r}_+ - \tilde{r})$.

The form of this loss function is motivated by previous work (Edunov et al., 2018; Shu et al., 2021), where they compare candidates among the model-generated ones. By contrast, we use CML to teach the model learn to prefer the corrected $\tilde{y}_+$ over the original $\tilde{y}$.

An astute reader may notice that the above formulation of CML does not *directly* use the gold label $y^*$. While this raises interesting possibilities of unsupervised variants, for the purpose of brevity we largely leave this to future work. Instead we can incorporate $y^*$ by also teaching the model to explicitly prefer $y^*$ over $\tilde{y}_+$ – since we would still like the model to ideally produce $y^*$ as the best candidate. Thus we add another CML term to define the following **corrective ranking loss**:

$$\text{CRL}_\theta(e') = \text{CML}_\theta(x, \tilde{y}, \tilde{r}, \tilde{y}_+, \tilde{r}_+) + \text{CML}_\theta(x, \tilde{y}_+, \tilde{r}_+, y^*, r^*). \tag{3}$$

To further prioritize the gold label we can still include NLL in our training loss as:

$$\lambda \times \text{NLL}_\theta(e) + (1.0 - \lambda) \times \text{CRL}_\theta(e'), \tag{4}$$

where $\lambda \in [0, 1.0]$ is a hyperparameter to control the regularization effect by $\text{NLL}_\theta(e)$. In practice we found that a small value of $\lambda$ helped stabilize learning and reduce variance (See Figure 6).

## 4 REWARD AND CORRECTION FUNCTIONS

### 4.1 REWARD FUNCTION

Practical applications for a generative model may span a variety of different complex reward functions which we would like to optimize for directly. Unfortunately public benchmarks do not come up with any such provided complex reward function. Thus we look to create a realistic reward function – for the running example of a sequence tagger / entity recognizer. Our reward function should be realistic enough to reflect the complexities of real-world metrics (and thus more complex than what traditional techniques can optimize) while still being understandable for us and readers.

As motivated in the introduction, practical application one may weight precision and recall differently. With $\beta$ controlling the importance of precision vs. recall, consider a reward of the form:

$$R(y, y^*) = \beta \times \text{precision}(y, y^*) + \text{recall}(y, y^*), \tag{5}$$

To further challenge the model, let us look into the definition of precision for sequence labeling – and highlight a practical issue that needs addressing. Consider this synthetic example:
 **- ground truth**:   $y^* = $ [A X] Y Z [B W U] [C V]
 **- prediction (1)**:   $y_1 = $ [D X] Y [E Z W U] [F V]  (a completely imprecise prediction)
 **- prediction (2)**:   $y_2 = $ X Y Z W U V     (an empty prediction)
The typical definition of precision would lead to $\text{precision}(y_1, y^*) = \text{precision}(y_2, y^*) = 0$, since the number of *true positives* is 0. However in practice these two predictions would have very different behaviors. When used to influence search engine behavior the first prediction would lead to far worse results (due to false positives) unlike the second prediction. This boils down to the precision metric not differentiating between false positives and empty predictions – which is a valid choice in sequential tagging problems.

To more closely reflect this distinction, let us modify our definition of the span-level precision to be:

$$\text{precision}'(y, y^*) = \frac{\text{TP} - c \times \text{FP}}{\text{TP} + \text{FP}}, \tag{6}$$

where TP and FP stand for true positive and false positive, respectively, and $c$ ($\geq 0$) is a hyperparameter to explicitly penalize false positives. When $c = 0$ we recover the classical precision metric. This small change now allows us to easily distinguish the above two predictions $y_1$ and $y_2$:

$$\text{precision}'(y_1, y^*) = \frac{0 - c \times 3}{0 + 3} = -c < \text{precision}'(y_2, y^*) = 0. \tag{7}$$

As an illustrative reward, we set $(c, \beta) = (2, 4)$ for our reward function in the rest of the paper – leading to reward scores between -8 (completely wrong) to +5 (perfect). We would like to reinforce that the above is just one example of a complex but realistic reward function. Experiments with other reward functions yielded similar – if not stronger – empirical findings (see Appendix G).

**Algorithm 1** Correction fn

```
 1: function C(y, y*)
 2:     yR = REPLACE(y, y*)
 3:     yD = DROP(yR, y*)
 4:     if yD == y* then
 5:         y+ = DROP(y, y*)
 6:     else
 7:         y+ = yD
 8:     end if
 9:     return y+
10: end function
```

| | Domain (Task) | Train | Valid | Test |
|---|---|---|---|---|
| ATIS | Travel (SF) | 4,478 | 500 | 893 |
| SNIPS | Assistant (SF) | 13,084 | 700 | 700 |
| MIT-R | Dining (SP) | 6,845 | 789 | 1,516 |
| MTOP en | Assistant (SP) | 15,667 | 2,235 | 4,386 |
| MTOP hi | Assistant (SP) | 11,330 | 2,012 | 2,789 |
| MTOP fr | | 11,814 | 1,577 | 3,193 |
| CoNLL03 | News (NER) | 14,987 | 3,466 | 3,684 |
| CoNLL00 | News (CHU) | 8,936 | 1,844 | 2,012 |

Table 2: Characteristics of datasets used in our experiments (SP=Semantic Parsing, SF = Slot Filling, NER = Named-Entity Recognition, CHU = Chunking).

### 4.2 CORRECTION FUNCTION

The goal of the correction function is to help explore the output space by yielding improved predictions $y_+ = C(y, y^*)$ such that $R(y_+, y^*) \geq R(y, y^*)$ w.h.p. While the suitability of a correction function may depend on the desired reward function, in this section we define three correction operations that we found widely effective across a slew of sequential tagging reward functions: **Drop**, **Replace** and **Annotate**.

**Drop:** Is designed to fix *precision* errors by dropping tagged spans in the prediction $y$ that do not match with the ground truth $y^*$. In addition to improving precision, this also often enables exploring new candidates – since NLL training does not explicitly teach the model to drop uncertain tags. For the example $y_1$ in Section 4.1, this would result in all predicted spans being dropped.

**Replace:** Works by replacing incorrect tags in $y$ with the correct ground-truth tags. For $y_1$ this would result in fixing "[D X]" and "[F V]" to produce "[A X] Y [E Z W U] [C V]".

**Annotate:** Improves recall by annotating untagged spans with the correct ground-truth tags.

We can also combine these operations. For example combining all three can lead to perfectly fixing any predictions. Given our example reward's focus on precision, we combine the first two (which improve precision) to give us our proposed correction function (see Algorithm 1). This function not only fixes most possible errors in predictions, but also helps efficiently explore the output space.

## 5 EXPERIMENTAL SETTINGS

### 5.1 DATASETS

For a comprehensive evaluation, we used six different datasets spanning different domains and tasks as detailed in Table 2. ATIS (Price, 1990) and SNIPS (Coucke et al., 2018) are two slot-filling tasks covering the travel (e.g., booking a flight ticket) and virtual assistant domains, respectively. MIT-restaurant (MIT-R)[1] and MTOP (Li et al., 2021) are semantic parsing datasets from the dining (e.g., ordering a dish) and voice assistant domains, respectively. The CoNLL 2003 dataset (Tjong Kim Sang & De Meulder, 2003) is from a news domain (i.e., Reuters) for the NER task. Lastly, the CoNLL 2000 dataset (Tjong Kim Sang & Buchholz, 2000) is also from a news domain (i.e., WSJ) for the syntactic chunking task. For space reasons, we report results for the latter two sets in the Appendix F. Additionally, to verify our findings hold across languages we also evaluated on the French and Hindi MTOP datasets with results are reported in Appendix A.

### 5.2 METHODS COMPARED

Our primary comparison is against the current standard – an model $NLL_\theta$ trained solely using the NLL loss. For the most fair and competitive baseline, we train the NLL model using the entire training set. This also forms the initialization point for all iterative models. Note that this means our iterative models do not benefit from any data specifically kept for iterative training.

While previous work (Raman et al., 2022) has shown that a generative NLL-based approach outperforms token-classification methods, for the sake of completeness we evaluate three additional

---

[1] https://groups.csail.mit.edu/sls/downloads/.

token classification baselines and report results in Appendix H. We also compare against the best existing (and most relevant) approach for reward optimization denoted as Shu++ (Shu et al., 2021). Shu++ relies solely on the decoder top-$k$ for exploring the reward space, and defines a contrastive loss between best- and worst-reward candidates; the comparison with Shu++ thus helps understand benefits of our correction function for better exploration in the reward optimization framework.

In addition to GROOT, we also evaluate a variant GROOT$_\text{NG}$ (NG: No Gold) which does not use the gold label in the corrective loss *i.e.* using $\text{CML}_\theta(x,\ \tilde{y},\ \tilde{r},\ \tilde{y}_+,\ \tilde{r}_+)$ instead of CRL (in Equation (4)).

## 5.3 MODEL

As an instance of a competitive sequence-to-sequence model, we use pre-trained mT5 (Xue et al., 2021) and build on the T5X code base (Roberts et al., 2022). We use a Base-sized model to maximize experimentation but verify results hold larger models (XXL-sized) in the Appendix.

We use the Adafactor optimizer (Shazeer & Stern, 2018) to train all models, along with Z-loss regularization (de Brébisson & Vincent, 2016). A constant learning rate of 0.001 is used for the NLL-only training stage, and 0.0001 otherwise.

NLL training is run for upto 2500 steps (evaluating checkpoints after every 100 steps), while iterative reward optimization runs for upto 200 steps (eval every 10 steps). For both NLL and iterative reward optimization we select the best checkpoint $\theta^{(t)}$ per the reward metric of the top-1 prediction on the validation set $\mathcal{D}^\text{val}$. Test set metrics are reported from the best checkpoint from the last iteration.

## 5.4 DEFAULT PARAMETER SETTINGS OF PROPOSED FRAMEWORK

Below are the default (and recommended) settings of our proposed framework. The effect of these different parameters are analyzed in Section 6.1 onwards and the Appendix.

- **Text format:** We use the "sentinel+tag (SI)" format to represent the input and output texts as recent studies (FitzGerald, 2020; Raman et al., 2022) have found that to be most effective and ideal to minimize hallucinations in text generation.
- **Data preparation:** We set the number of reward optimization iterations as $T = 9$ (Section 3.2) and create $\mathcal{L}_t$ from the entire training set at each iteration.
- **Beam size:** We set $k = 5$ as a beam size for the top-$k$ prediction stage in both training and evaluation.
- **Candidate selection:** To select the candidate to improve $\tilde{y}$ from the top-k, we default to selecting the highest reward candidate.
- **Candidate correction:** We use the correction function described in Section 4.2.
- **Loss function:** We set $(\alpha, \lambda) = (0.5, 0.1)$ for our loss function in Section 3.3.

**Note**: To avoid any "*human overfitting*", all development of the methodology (incl. refinements and ablations) was conducted using **only the validation sets**. Test sets were only used to report final results for the paper (in a completely human-blind fashion).

## 5.5 EVALUATION METRICS

We evaluate each example by the following four metrics, and report macro averaged scores. All the reported scores are based on mean and standard deviation across five different runs. Results for additional metrics are reported in the Appendix.

- **Top-1 reward:** Equation (5) for the top-1 candidate by the beam search. (Maximum value: 5.0)

- **Average reward:** Reward averaged across the top-$k$ candidates. (Maximum value: 5.0)

- **Max reward:** The highest reward value among the top-$k$ candidates. (Maximum value: 5.0)

- **Rank correlation:** Spearman's rank correlation coefficient to measure if the model has correctly predicted the top-k candidates in order of their reward values. (Maximum value: 1.0)

## 6 RESULTS AND DISCUSSIONS

As seen from the test results in Tables 3 and 8, both GROOT variants significantly outperform the NLL and Shu++ baselines across (nearly) all metrics on all six datasets. GROOT not only improves

| | ATIS | | | | SNIPS | | | | mTOP (en) | | | | MIT-R | | | |
|---|---|---|---|---|---|---|---|---|---|---|---|---|---|---|---|---|
| | NLL | Shu++ | GROOT | GROOT$_{NG}$ | NLL | Shu++ | GROOT | GROOT$_{NG}$ | NLL | Shu++ | GROOT | GROOT$_{NG}$ | NLL | Shu++ | GROOT | GROOT$_{NG}$ |
| Top-1 reward | 4.257$^\dagger$ ± 0.030 | 4.038 ± 0.130 | **4.333$^\dagger$** ± 0.028 | 4.300$^\dagger$ ± 0.033 | 4.066 ± 0.185 | 4.058 ± 0.052 | **4.231$^\dagger$** ± 0.074 | 4.215$^\dagger$ ± 0.029 | 3.815 ± 0.068 | 3.835 ± 0.018 | **4.006$^\dagger$** ± 0.036 | **4.039$^\dagger$** ± 0.001 | 2.467 ± 0.131 | **2.614** ± 0.011 | 2.852$^\dagger$ ± 0.074 | **3.190$^\dagger$** ± 0.020 |
| Average reward | 0.746 ± 0.115 | 0.105 ± 1.876 | 2.737$^\dagger$ ± 0.290 | **2.797$^\dagger$** ± 0.114 | -0.015 ± 0.146 | **0.679** ± 0.058 | 2.089$^\dagger$ ± 0.124 | **2.159$^\dagger$** ± 0.144 | -0.926 ± 0.112 | **0.140** ± 0.273 | 0.208 ± 0.068 | **1.145$^\dagger$** ± 0.165 | -0.523 ± 0.110 | 0.502 ± 0.145 | 0.659 ± 0.106 | **0.838$^\dagger$** ± 0.020 |
| Max reward | 4.630$^\dagger$ ± 0.032 | 4.375 ± 0.184 | **4.792$^\dagger$** ± 0.021 | **4.759$^\dagger$** ± 0.029 | 4.831$^\dagger$ ± 0.054 | 4.720 ± 0.043 | **4.922$^\dagger$** ± 0.013 | 4.911$^\dagger$ ± 0.010 | 4.719$^\dagger$ ± 0.031 | 4.643 ± 0.034 | **4.847$^\dagger$** ± 0.013 | 4.827$^\dagger$ ± 0.031 | 4.297$^\dagger$ ± 0.076 | 4.057 ± 0.016 | 4.607$^\dagger$ ± 0.051 | **4.679$^\dagger$** ± 0.011 |
| Rank correlation | 0.677 ±0.008 | **0.688** ±0.005 | **0.800$^\dagger$** ±0.092 | 0.759 ±0.073 | 0.714$^\dagger$ ±0.010 | 0.659 ±0.049 | **0.837$^\dagger$** ±0.023 | 0.826$^\dagger$ ±0.019 | 0.694$^\dagger$ ±0.005 | 0.665 ±0.009 | **0.803$^\dagger$** ±0.006 | 0.740$^\dagger$ ±0.011 | 0.629 ±0.007 | 0.651 ±0.011 | 0.656 ±0.004 | **0.658** ±0.007 |

Table 3: Test set results on the four datasets. Standard deviation values are over 5 runs. **Bolded** values and the $^\dagger$ symbol indicate 99% significance (via z-test) vs. NLL and Shu++ respectively.

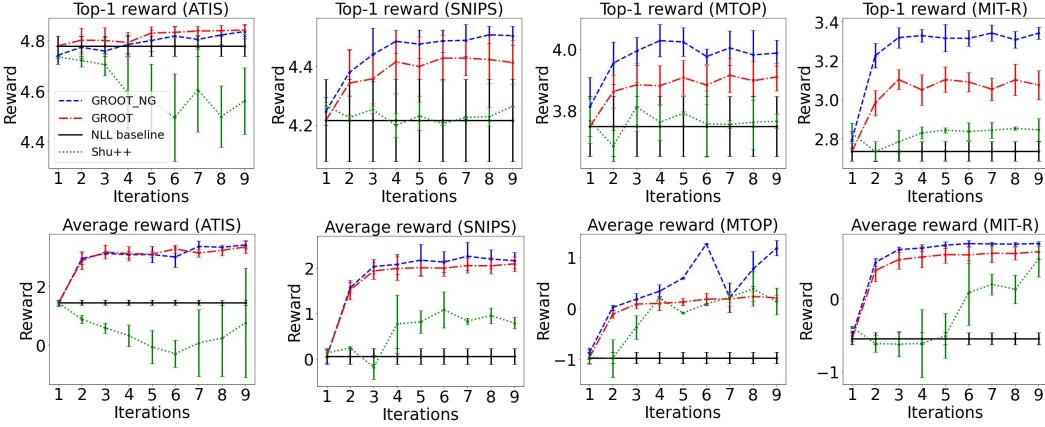

Figure 3: Performance of different loss functions on the top-1 and average reward metrics. For all figures (here and below) error bars denote standard deviations across 5 runs.

the reward of the top-1 prediction consistently, but also significantly improves the quality of the top-$k$ predictions as evidenced by the large increases in the Average Reward scores. GROOT methods also score highest when looking at the highest reward scores among the top-$k$. The improved rank correlations provide the final confirmation that the GROOT models do indeed gain a better understanding of the output space in regards to the provided (blackbox) reward function.

These results also demonstrate the importance of efficiently exploring the output space. As seen in Table 3, the Shu++ model is often no better – and in many cases significantly worse – than the NLL baseline. This clearly illustrates that simply relying on the top-$k$ may not suffice. More specifically the lower quality of the top-$k$ of the NLL baseline provides very little room for hill-climbing towards better reward scores, which in turn leads to the poor scores observed. In contrast the GROOT methods outperform Shu++ on every metric, proving that the correction-function based exploration coupled with the contrastive loss functions leads to a significantly better model.

## 6.1 EFFECT OF LOSS FUNCTIONS

Figure 3 shows how the top-1 and average reward values change over the course of the iterative training process. We can clearly see that GROOT methods smoothly improve the reward scores and improve the decoder output distribution. Between the GROOT variants, we find that GROOT$_{NG}$ tends to more aggressively optimize the intended reward (since it is not anchored to also optimize the gold label probability). This does come up with a drawback as other metrics different than the provided reward may degrade (as shown in Table 7 in Appendix), due to the focus on the provided reward. Regardless the strong performance of GROOT$_{NG}$ hints at the promise of such techniques even being used in low supervision / unsupervised settings. However we leave this to be explored in future work and instead focus on the vanilla GROOT model in the rest of the analyses.

## 6.2 EFFECT OF CORRECTION FUNCTION

Figure 4 shows the effect of the correction function used. While "Drop+replacement" corresponds to our default correction function (Algorithm 1), "Only drop" only uses the drop operation. We can see that the drop operation by itself is quite effective, due to the increased exploration it offers relative to NLL-only training.

Coupled with the earlier results, we clearly see how correction functions can help break the limits of exploration in Seq2Seq models. While alternatives such as decoder distribution sampling (Ranzato et al., 2016; Ackley et al., 1985; Ficler & Goldberg, 2017) do exist, the highly skewed output distributions of the NLL model (as seen in Table 1) often constrain exploration and need significant

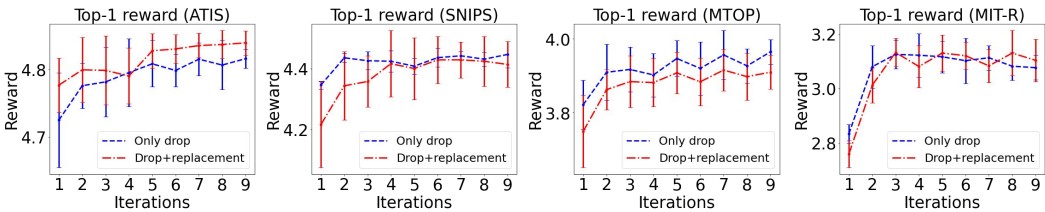

Figure 4: Comparison between different correction functions.

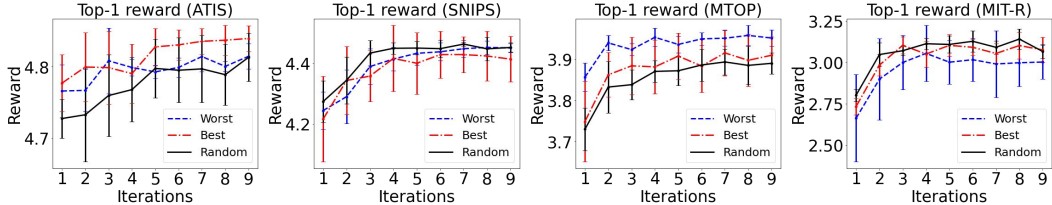

Figure 5: Comparison between different candidate selection strategies.

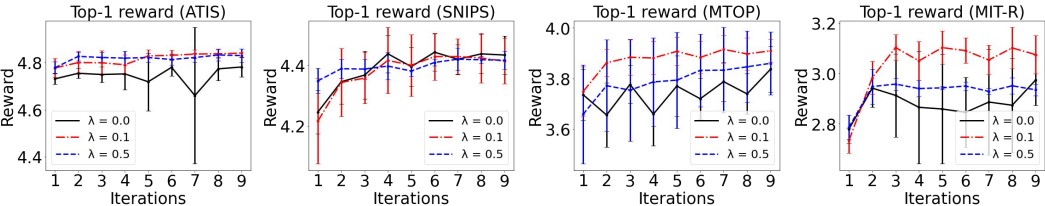

Figure 6: Comparison between different values of $\lambda$ in Equation (4).

tuning of temperature parameters to smooth out the distribution. Thus we conclude that correction functions provide a far simpler and more effective way of exploring the output space.

### 6.3 Effect of Candidate Selection Strategy

Figure 5 shows how different candidate selection strategies affect the results. While our default choice corresponding to "Best" is often the most consistent, other strategies such as "Worst" (lowest reward candidate) or "Random" are also effective at helping the model learn reward contours. Thus we believe that our framework is not sensitive to the selection strategies, and could even incorporate multiple candidates via different ranking objectives (Jagerman et al., 2022).

### 6.4 Effect of $\lambda$

Figure 6 shows how the NLL loss contribution (controlled by $\lambda$ in Equation 4) affects results. In general we find that a small contribution of NLL – as seen in the $\lambda = 0.1$ curves – generally help to stabilize learning, while larger values still do benefit from iterative improvements of reward (albeit slightly less). $\lambda = 0.0$ though was fairly unstable training, and thus we recommend setting $\lambda > 0$.

## 7 Conclusion

We have presented a simple yet effective framework, GROOT, and empirically verified the effectiveness on sequential labeling tasks across multiple tasks and languages. Via the use of a correction function based exploration and a contrastive loss, GROOT is able to directly and effectively optimize a Seq2Seq model for the given blackbox reward function. Furthermore, unlike standard maximum likelihood training, the output distribution of GROOT also significantly improves and is well correlated with the desired reward metric.

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

APPENDIX

## A    RESULTS IN NON-ENGLISH LANGUAGES

Table 4 shows results on the MTOP dataset in Hindi and French. The effectiveness of our method is consistently observed in Hindi and French as well.

|  | MTOP (hi) | | MTOP (fr) | |
| --- | --- | --- | --- | --- |
|  | NLL baseline | GROOT | NLL baseline | GROOT |
| Top-1 reward | $3.198 \pm 0.076$ | $3.411 \pm 0.077$ | $3.419 \pm 0.049$ | $3.663 \pm 0.041$ |
| Average reward | $-1.524 \pm 0.068$ | $-0.440 \pm 0.125$ | $-1.594 \pm 0.031$ | $-0.377 \pm 0.047$ |
| Max reward | $4.454 \pm 0.029$ | $4.736 \pm 0.019$ | $4.551 \pm 0.017$ | $4.796 \pm 0.017$ |
| Rank correlation | $0.688 \pm 0.003$ | $0.768 \pm 0.012$ | $0.713 \pm 0.006$ | $0.800 \pm 0.002$ |

Table 4: Test set results on the Hindi and French MTOP dataset.

## B    RESULTS WITH MT5 XXL

We mainly used mT5 BASE[2] to maximize experimentation and analyze the effectiveness of different aspects of our proposed framework. While this is already a large transformer model (Vaswani et al., 2017), a natural question one may ask is whether the observed gains carry over to even larger models.

To answer this question, we verify the effectiveness using mT5 XXL.[3] With 13B+ parameters, the XXL variant is significantly larger than mT5 BASE as described in Xue et al. (2021). For example, mT5 BASE consists of 12 transformer layers with 768-dimensional embeddings and 2048-dimensional Multi-Layer Perceptrons (MLPs), while mT5 XXL consists of 24 layers with 4096-dimensional embeddings and 10240-dimensional MLPs.

Table 5 shows the results. When compared with the results in Table 3, it is unsurprising to see significant gains for all methods using mT5 XXL over mT5 BASE. However we still observe the same trends with GROOT improving reward-based metrics. It is interesting to note that while the average reward scores improve significantly by simply using the larger model (except on ATIS), GROOT still helps maintain a sizeable gap. Thus we believe our reward optimization framework will help even when larger and larger models are invented in the future.

|  | ATIS | | SNIPS | | MTOP (en) | | MIT-R | |
| --- | --- | --- | --- | --- | --- | --- | --- | --- |
|  | NLL | GROOT | NLL | GROOT | NLL | GROOT | NLL | GROOT |
| Top-1 reward | $4.310 \pm 0.028$ | $4.360 \pm 0.027$ | $4.398 \pm 0.062$ | $4.482 \pm 0.057$ | $4.136 \pm 0.059$ | $4.175 \pm 0.101$ | $2.592 \pm 0.169$ | $2.926 \pm 0.117$ |
| Average reward | $0.550 \pm 0.180$ | $2.628 \pm 0.089$ | $0.721 \pm 0.202$ | $2.576 \pm 0.213$ | $0.005 \pm 0.202$ | $0.609 \pm 0.165$ | $-0.380 \pm 0.124$ | $0.716 \pm 0.124$ |
| Max reward | $4.616 \pm 0.022$ | $4.768 \pm 0.018$ | $4.857 \pm 0.013$ | $4.902 \pm 0.033$ | $4.764 \pm 0.032$ | $4.865 \pm 0.016$ | $4.403 \pm 0.099$ | $4.629 \pm 0.042$ |
| Rank correlation | $0.644 \pm 0.028$ | $0.698 \pm 0.105$ | $0.629 \pm 0.011$ | $0.849 \pm 0.028$ | $0.634 \pm 0.006$ | $0.785 \pm 0.047$ | $0.621 \pm 0.007$ | $0.655 \pm 0.020$ |

Table 5: Test set results with mT5 XXL.

## C    EXAMPLE PREDICTIONS

Table 6 shows how the prediction examples in Table 1 (a) are modified by GROOT. If the model is not confident enough about the tag of "moorpark," the top-1 prediction leaves it untagged.

| NLL | Model predictions for an example from the SNIPS *validation* set |
| --- | --- |
| 0.154 | book [party_size_number seven] in [spatial_relation neighboring] moorpark |
| 0.712 | book [party_size_number seven] in [spatial_relation neighboring] **[geographic_poi moorpark]** |
| 1.303 | book [party_size_number seven] in [spatial_relation neighboring] **[poi moorpark]** |
| 1.383 | book [party_size_number seven] in [spatial_relation neighboring] [city moorpark] (*) |
| 2.004 | book [party_size_number seven] in [spatial_relation neighboring] **[object_name moorpark]** |

Table 6: Top-5 predictions refined by GROOT. This can be contrasted with Table 1 (a).

---

[2] https://github.com/google-research/t5x/blob/main/t5x/examples/t5/mt5/base.gin.

[3] https://github.com/google-research/t5x/blob/main/t5x/examples/t5/mt5/xxl.gin.

# D    ADDITIONAL EVALUATION METRICS

In addition to the evaluation metrics introduced in Section 5.5, we also report the results with the following metrics for reference:

**- Top-1 precision′:** Equation (6) for the top-1 candidate. (Maximum value: 1.0)

**- Top-1 precision:** The standard precision metric for the top-1 candidate. (Maximum value: 1.0)

**- Top-1 recall:** The standard recall metric for the top-1 candidate. (Maximum value: 1.0)

**- Top-1 EM:** A binary score to check if the top-1 candidate perfectly matches the ground-truth annotation. (Maximum value: 1.0)

Table 7 shows the results, corresponding to those in Table 3. As expected, the precision′ scores are significantly improved, while slightly degrading recall (and hence EM).

| | ATIS | | | | SNIPS | | | | MTOP (en) | | | | MIT-R | | | |
| --- | --- | --- | --- | --- | --- | --- | --- | --- | --- | --- | --- | --- | --- | --- | --- | --- |
| | NLL | GROOT | GROOT$_{NG}$ | Shu++ | NLL | GROOT | GROOT$_{NG}$ | Shu++ | NLL | GROOT | GROOT$_{NG}$ | Shu++ | NLL | GROOT | GROOT$_{NG}$ | Shu++ |
| Top-1 precision′ | 0.829 | 0.849 | 0.841 | 0.777 | 0.783 | 0.826 | 0.836 | 0.781 | 0.726 | 0.776 | 0.796 | 0.732 | 0.413 | 0.521 | 0.633 | 0.447 |
| | ± 0.007 | ± 0.008 | ± 0.007 | ± 0.031 | ± 0.043± | ± 0.015 | ± 0.007 | ± 0.012 | ± 0.015 | ± 0.007 | ± 0.007 | ± 0.002 | ± 0.031 | ± 0.022 | ± 0.013 | ± 0.002 |
| Top-1 precision | 0.942 | 0.943 | 0.939 | 0.925 | 0.928 | 0.934 | 0.908 | 0.927 | 0.906 | 0.912 | 0.893 | 0.905 | 0.803 | 0.821 | 0.809 | 0.815 |
| | ± 0.003 | ± 0.003 | ± 0.006 | ± 0.011 | ± 0.014 | ± 0.013 | ± 0.002 | ± 0.004 | ± 0.006 | ± 0.005 | ± 0.012 | ± 0.003 | ± 0.010 | ± 0.004 | ± 0.016 | ± 0.001 |
| Top-1 recall | 0.942 | 0.938 | 0.935 | 0.930 | 0.934 | 0.927 | 0.871 | 0.934 | 0.913 | 0.900 | 0.854 | 0.907 | 0.814 | 0.766 | 0.658 | 0.825 |
| | ± 0.003 | ± 0.006 | ± 0.004 | ± 0.007 | ± 0.014 | ± 0.016 | ± 0.002 | ± 0.003 | ± 0.006 | ± 0.009 | ± 0.027 | ± 0.011 | ± 0.009 | ± 0.015 | ± 0.031 | ± 0.004 |
| Top-1 EM | 0.897 | 0.891 | 0.889 | 0.866 | 0.869 | 0.871 | 0.787 | 0.865 | 0.838 | 0.838 | 0.775 | 0.836 | 0.623 | 0.583 | 0.447 | 0.635 |
| | ± 0.005 | ± 0.010 | ± 0.004 | ± 0.021 | ± 0.023 | ± 0.027 | ± 0.011 | ± 0.008 | ± 0.010 | ± 0.012 | ± 0.033 | ± 0.010 | ± 0.015 | ± 0.015 | ± 0.033 | ± 0.003 |

Table 7: Additional evaluation metrics for the results in Table 3.

# E    MORE ANALYSIS RESULTS

Figures 7, 8, 9, and 11 show how the average reward scores are improved in the course of the iterative reward optimization. The average reward scores are smoothly improved in all the cases. We can also see similar trends that are observed in the cases of the top-1 reward metric.

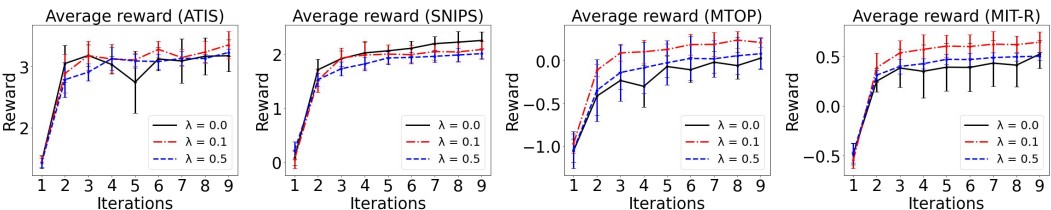

Figure 7: Comparison between different values of $\lambda$ in Equation (4) for the average reward metric.

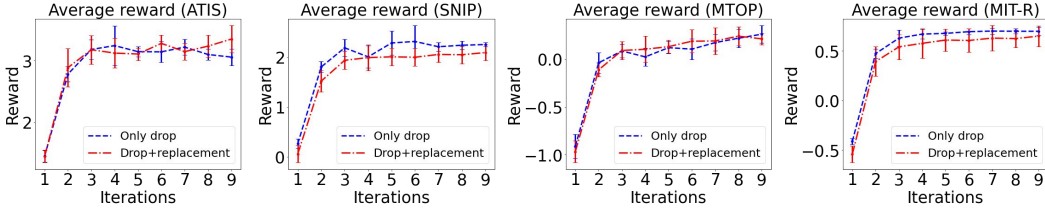

Figure 8: Comparison between different correction functions for the average reward metric.

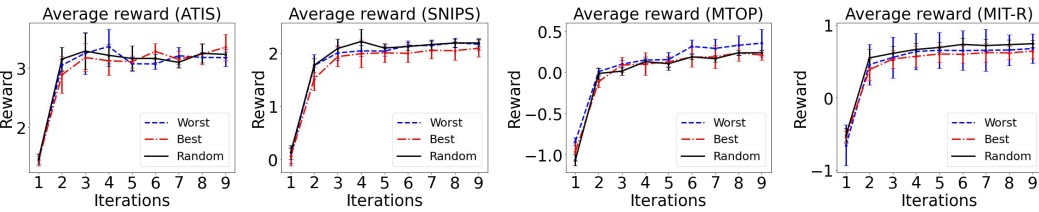

Figure 9: Comparison between different candidate selection strategies for the average reward metric.

### E.1 DOES A MORE TEST-LIKE DATA SPLIT WORK?

As mentioned in Section 5.2 we use "all" of the training examples for NLL and iterative training stages. However as seen in Table 1, NLL training tends to overfit heavily leading to very skewed distributions. Instead a more test-like split would use only part of the data for NLL and then a new batch for the first stage of training and so on. This setup more closely simulates test-time behavior since the model would not have seen any of the examples before. However given the limited training set size such splitting may seem likely to degrade performance. Thus we compared against a variant where the training set was "split" into 3 pieces which are cyclically used for each iteration.

As seen in Figure 10 despite starting from a much worse initialization, the GROOT model quickly recovers outperforming NLL and nearly catching upto the "all" performance on all datasets. For reference, we also show Figure 11. We believe with a more careful data strategy and/or a larger training set, we could see further gains.

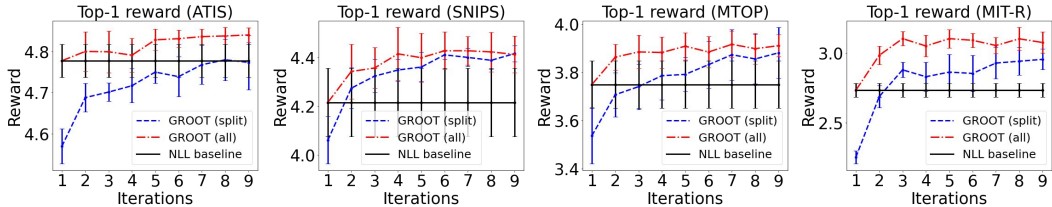

Figure 10: Comparison between different data preparation strategies.

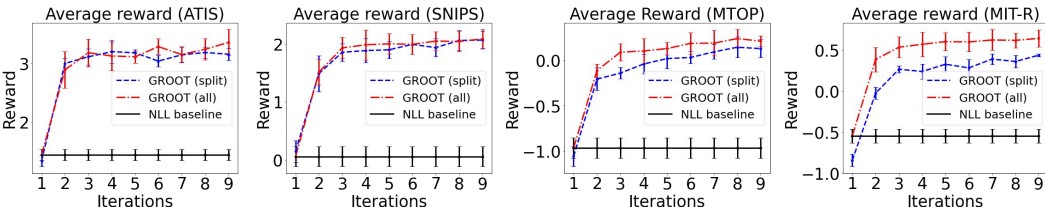

Figure 11: Comparison between different data preparation strategies for the average reward metric.

## F EXPERIMENTS ON DIFFERENT TASKS

This section shows how GROOT works on the NER and chunking tasks, where all the experimental settings are consistent with those on the ATIS, SNIPS, mTOP, and MIT-R datasets. Table 8 shows the results. We can see that GROOT consistently improves upon the NLL baseline in all the reward-based metrics on the two tasks, while Shu++ does not. Those results show the generalization ability of our proposed framework across different tasks.

| | CoNLL03 (NER) | | | CoNLL00 (Chunking) | | |
| | NLL | Shu++ | GROOT | NLL | Shu++ | GROOT |
|---|---|---|---|---|---|---|
| Top-1 reward | $3.593 \pm 0.193$ | $3.670 \pm 0.070$ | $3.720 \pm 0.108$ | $4.469 \pm 0.057$ | $4.423 \pm 0.048$ | $4.500 \pm 0.035$ |
| Average reward | $-1.837 \pm 0.171$ | $0.581 \pm 1.057$ | $-0.045 \pm 0.507$ | $3.357 \pm 0.051$ | $2.261 \pm 0.151$ | $4.131 \pm 0.055$ |
| Max reward | $4.645 \pm 0.059$ | $4.352 \pm 0.121$ | $4.796 \pm 0.038$ | $4.842 \pm 0.016$ | $4.754 \pm 0.034$ | $4.865 \pm 0.011$ |
| Rank correlation | $0.695 \pm 0.012$ | $0.707 \pm 0.076$ | $0.786 \pm 0.020$ | $0.543 \pm 0.003$ | $0.600 \pm 0.019$ | $0.642 \pm 0.013$ |

Table 8: Test set results on the NER and chunking tasks.

## G EXPERIMENTS WITH DIFFERENT REWARD FUNCTIONS

As discussed in Section 4.1, we limited our exposition in the main body of the paper to a single reward function for simplicity. However we found similar – if not stronger – empirical results with other reward functions as described in this section.

### G.1 DEFINITIONS: VARIANTS OF REWARD FUNCTIONS

Our default reward function defined by Equations (5) and (6) – which we denote as $R_{\text{PR}}$ linearly interpolated a variant of precision and recall. This reward function inherently favors a small number of highly confident spans. The results in Table 7 show such a tradeoff. To explore a different set of rewards with potentially different characteristics we took inspiration from the popular Jaccard similarity (Jaccard, 1912) and the Tversky index (Tversky, 1977).

#### G.1.1 MODIFIED JACCARD SIMILARITY

The Jaccard similarity can be defined as:

$$\text{JAC}(y, y^*) = \frac{|Y \cap Y^*|}{|Y \cup Y^*|} = \frac{\text{TP}}{\text{TP} + \text{FP} + \text{FN}}, \tag{8}$$

where $Y$ is the set of the predicted spans, $Y^*$ is the set of the ground-truth spans, and FN stands for false negative. Compared with $\text{precision}'$ in Equation (6), this already takes recall into account by false negatives.

As another example of a more complex reward function, we can extend this with a similar penalty coefficient $c$ to obtain:

$$R_{\text{JAC}}(y, y^*) = \frac{|Y \cap Y^*| - c \times |Y \setminus Y^*|}{|Y \cup Y^*|} = \frac{\text{TP} - c \times \text{FP}}{\text{TP} + \text{FP} + \text{FN}}, \tag{9}$$

#### G.1.2 MODIFIED TVERSKY INDEX

The Tversky index is a generalization of the Jaccard similarity and Sorensen-Dice coefficient (Sorensen, 1948; Dice, 1945) which allows for fine-grained control of the balance between different error classes:

$$\text{TVE}(y, y^*) = \frac{|Y \cap Y^*|}{|Y \cap Y^*| + \gamma \times |Y \setminus Y^*| + \omega \times |Y^* \setminus Y|} = \frac{\text{TP}}{\text{TP} + \gamma \times \text{FP} + \omega \times \text{FN}}, \tag{10}$$

where $\gamma$ and $\omega$ are hyperparameters. We can further extend this in an analogous manner as above to define the following reward function:

$$R_{\text{TVE}}(y, y^*) = \frac{|Y \cap Y^*| - c \times |Y \setminus Y^*|}{|Y \cap Y^*| + \gamma \times |Y \setminus Y^*| + \omega \times |Y^* \setminus Y|} = \frac{\text{TP} - c \times \text{FP}}{\text{TP} + \gamma \times \text{FP} + \omega \times \text{FN}}. \tag{11}$$

### G.2 SETTINGS

We use the MTOP (en) and MIT-R datasets to verify that GROOT effectively works with these reward functions. We simply replace the original reward function with the new reward functions and did not change any other settings including the hyperparameters. The only change we change we make is to re-scale the value of $R_{\text{JAC}}$ and $R_{\text{TVE}}$ by a factor of $(\beta + 1.0)$ in the loss function to align the value ranges of rewards similar to what we had previously.

We report evaluation scores for the associated reward function that was used for training in each setup. For example, we use $R_{\text{JAC}}$ for evaluation if $R_{\text{JAC}}$ is used for training. We report results (averaged over 5 runs) for $\{R_{\text{JAC}}, c = 1.0\}$ and $\{R_{\text{TVE}}, c = 1.0, \gamma = 0.5, \omega = 1.0\}$. We note that we have observed consistent results with other setups.

### G.3 RESULTS

Tables 9 and 10 show the results. We can see that GROOT consistently improves upon the NLL baseline in all the reward-based metrics, while Shu++ does not. These results empirically verify the robustness of our proposed reward optimization framework.

## H TOKEN CLASSIFICATION RESULTS

Previous works (Raman et al., 2022) has shown that a generative NLL-based approach outperforms other token-classification approaches. However, for the sake of completeness we implemented and evaluated three additional baselines:

|  | MTOP (en) | | | MIT-R | | |
|---|---|---|---|---|---|---|
|  | NLL | Shu++ | GROOT | NLL | Shu++ | GROOT |
| Top-1 reward | $0.836 \pm 0.012$ | $0.845 \pm 0.001$ | $0.837 \pm 0.009$ | $0.621 \pm 0.027$ | $0.618 \pm 0.014$ | $0.651 \pm 0.005$ |
| Average reward | $0.133 \pm 0.013$ | $0.329 \pm 0.033$ | $0.200 \pm 0.005$ | $0.172 \pm 0.019$ | $0.100 \pm 0.062$ | $0.263 \pm 0.010$ |
| Max reward | $0.958 \pm 0.003$ | $0.947 \pm 0.004$ | $0.963 \pm 0.002$ | $0.884 \pm 0.162$ | $0.860 \pm 0.005$ | $0.890 \pm 0.006$ |
| Rank correlation | $0.701 \pm 0.009$ | $0.708 \pm 0.002$ | $0.811 \pm 0.017$ | $0.614 \pm 0.006$ | $0.646 \pm 0.020$ | $0.673 \pm 0.004$ |

Table 9: Test set results with $R_{\text{JAC}}$ on the MTOP (en) and MIT-R datasets.

|  | MTOP (en) | | | MIT-R | | |
|---|---|---|---|---|---|---|
|  | NLL | Shu++ | GROOT | NLL | Shu++ | GROOT |
| Top-1 reward | $0.815 \pm 0.011$ | $0.816 \pm 0.003$ | $0.835 \pm 0.007$ | $0.615 \pm 0.018$ | $0.605 \pm 0.018$ | $0.634 \pm 0.008$ |
| Average reward | $0.048 \pm 0.017$ | $0.249 \pm 0.027$ | $0.140 \pm 0.017$ | $0.147 \pm 0.012$ | $0.095 \pm 0.034$ | $0.239 \pm 0.005$ |
| Max reward | $0.953 \pm 0.003$ | $0.937 \pm 0.003$ | $0.965 \pm 0.002$ | $0.887 \pm 0.007$ | $0.862 \pm 0.009$ | $0.891 \pm 0.002$ |
| Rank correlation | $0.700 \pm 0.002$ | $0.679 \pm 0.003$ | $0.818 \pm 0.006$ | $0.614 \pm 0.008$ | $0.645 \pm 0.017$ | $0.666 \pm 0.007$ |

Table 10: Test set results with $R_{\text{TVE}}$ on the MTOP (en) and MIT-R datasets.

- **mBERT**: This uses an mBERT (encoder-only) model (Devlin et al., 2018) to predict the per-token labels. We used the Base-sized mBERT model as the most comparable with the Base-sized mT5 backbone used in most of our experiments.

- **mT5**: As shown in prior work (Lewis et al., 2019; Raman et al., 2022), generative models (like T5) can also be used to predict per-token labels. Thus we used the same mT5-Base model to predict the sequence of BIO token labels. As recommended by Raman et al. (2022), we use a single *Inside* label – rather than a per-class label – to improve performance.

- **mT5 (+Input)**: The results from Raman et al. (2022) showed that learning to generate the input tokens **along with** the BIO token labels, leads to better performance using mT5 models. Thus we evaluated this as an additional baseline.

The results for these additional baselines are provided in Table 11. When compared with the other methods (Table 3 for $R_{\text{PR}}$, Table 9 for $R_{\text{JAC}}$ and Table 10 for $R_{\text{TVE}}$), we find that the GROOT significantly outperforms all three token classification approaches – a further validation of our approach. Note that GROOT may potentially be helpful in the token classification setting as well. However we leave this for future work to explore, given the general potency of the generative approach.

|  | ATIS | | | SNIPS | | | mTOP(en) | | | MIT-R | | |
|---|---|---|---|---|---|---|---|---|---|---|---|---|
|  | mBERT | mT5 | mT5 (+Input) | mBERT | mT5 | mT5 (+Input) | mBERT | mT5 | mT5 (+Input) | mBERT | mT5 | mT5 (+Input) |
| $R_{\text{PR}}$ | 4.0 | 4.09 | 4.17 | 4.11 | 4.16 | 4.09 | 3.31 | 3.46 | 3.63 | 1.37 | 2.37 | 2.52 |
| $R_{\text{JAC}}$ | 0.859 | 0.878 | 0.887 | 0.876 | 0.884 | 0.874 | 0.741 | 0.762 | 0.785 | 0.479 | 0.612 | 0.629 |
| $R_{\text{TVE}}$ | 0.854 | 0.871 | 0.882 | 0.872 | 0.879 | 0.87 | 0.725 | 0.746 | 0.771 | 0.456 | 0.599 | 0.618 |

Table 11: Test set results for the top-1 rewards of different token-classification baselines (results averaged over three runs).

