# OpenReview forum: "GROOT: Corrective Reward Optimization for Generative Sequential Labeling"
_ICLR.cc/2023/Conference — Submitted to ICLR 2023_

### Official Review · Reviewer_oMZW · 2022-10-23

**Confidence:** 3
**Correctness:** 2
**Technical Novelty And Significance:** 3
**Empirical Novelty And Significance:** 3
**Recommendation:** 5

**Clarity, Quality, Novelty And Reproducibility:**

The paper has clear motivation and well exposes their method. The empirical results as well as the ablation studies show convincing results for their approach. The way to explore more candidates seem normal to me, although the proposed loss has no particular novelty. The margion loss needs to be adjusted in the approach so potentially hinders reproducibility; nevertheless, the approach is interesting and does seem to solve some practical issues.

**Strength And Weaknesses:**

Strength:
- The paper exposes a clear motivation and proposes novel correction function to better explore the space.
- The paper shows strong empirical results
- The paper has done quite thorough ablation studies showing the effectivenss of their methods as well being stable and not very senstible to various settings

Weakness:
- The paper exposes results in one particular reward function setting (section 4.1) which shows convincing results. However, to know that the proposed algorithm is effective, one should at least know the performance of other settings; it is mentioned in the paper that other settings yielded similar empirical findings, however I can not find those findings including appendix. Without this, I can't know if the paper has proposed an effective algorithm or not.
- (Question) I don't' understand why the paper proposes to incorporate these methods only once the model is fined tuned. The method is general and can be applied as a fine tuning method. Not sure why there is no experiment as well as no mentioning on this.


**Summary Of The Paper:**

Seq2seq based is a popular approach for generative sequence labelling. However, the popoular log likelihood training doesn't always model well the loss each practical problem speicfies.

In this paper, the authors first pose such a setting, and then proposes a framework (GROOT) that allows to incoporate those information in generative sequence modelling frameworks. Authors propose two novel techniques to solve this issue: 1) correction function which incrementally improves over the current predicion model 2) Margin loss that can be applied to leverage current predicitons, gold annotations as well as the candidates generated by correction function. Through empirical studies, the authors show that their method improves over baseline as well as recently proposed methods.

**Summary Of The Review:**

For generative sequence labelling, authors propose to model reward to better guide seq2seq learning. Authors propose two novel techniques to solve this issue: 1) correction function which incrementally improves over the current predicion model 2) Margin loss that can be applied to leverage current predicitons, gold annotations as well as the candidates generated by correction function. Through empirical studies, the authors show that their method improves over baseline as well as recently proposed methods.

My main concern is that I can't find support that the algorithms authors propose generalize to more scenarios than the ones posed in 4.1.

---

> ### Author Response · Authors · 2022-11-20
> **Response to the comments**
>
> Thank you for taking the time to read our paper and sharing your insightful feedback. You have our sincere thanks and gratitude.
>
> Regarding your questions and comments, please see below:
>
> **Experiments with other reward functions**:
>
> Thank you for pointing out this. This is indeed a fair concern. To address this, we have added results for a couple of different reward functions to demonstrate the effectiveness of our method as much as possible as seen in Appendix G.
>
> We tried to cast a broad net with these new reward functions – while keeping them simple enough to explain and understand. We hope this addresses your concern regarding the generality of the technique. If you have suggestions for alternative reward functions please let us know and we would be happy to include results for them (On yet-unreleased data we have tried a broad variety of reward functions – including model-based rewards – and found consistent results, and hence we feel confident the actual formulation of the reward function is not crucial to performance of the method.)
>
> **Regarding margin scaling and loss**:
>
> You are right to wonder if the margin-loss needs to be adjusted based on the reward function. However as noted in Appendix G, we have found that naively scaling the reward with no tuning works out-of-the-box for all the reward functions /  tasks / model sizes we experimented with.
>
> Regarding the loss, while you are correct in that a margin-based contrastive loss is not new to ML/NLP, we’d like to believe that its use in training generative models coupled with a smoother hill-climbing using a correction function is a novel change.
>
> **Question regarding the fine-tuning strategy**:
>
> Thank you for asking this question. To clarify why the model needs some initial finetuning with (potentially minimal) human-labeled data, consider what happens without it.
>
> Without this initial fine-tuning step, a randomly initialized (or a pre-trained) model cannot generate meaningful predictions for a particular task. This would result in a fairly uninformative top-k predictions – which would make hill-climbing on the rewards significantly harder. For this reason, the initial seed training with NLL is necessary.
>
> A different way to answer your question would be to understand what happens when the initial finetuning uses minimal data: This is indeed a very interesting question and a direction we are actively pursuing. Our results in Appendix E.1 (previously Section 6.5) demonstrate that even if we started with a much smaller finetuning set, we still can improve upon NLL in all cases. We have ongoing work that perhaps can further these gains as well – which we leave to future work.
>
> We hope the above changes (coupled with the other experimental results and clarifications we’ve added) will help address the main concerns you had about our work. Please let us know if you have any further questions.

---

### Official Review · Reviewer_PagF · 2022-10-25

**Confidence:** 4
**Correctness:** 2
**Technical Novelty And Significance:** 4
**Empirical Novelty And Significance:** Not applicable
**Recommendation:** 5

**Clarity, Quality, Novelty And Reproducibility:**

Quality: 3/5
Clarity: 5/5
Originality: 5/5


**Strength And Weaknesses:**

Strengths:
* The paper is well-written and easy to understand.
* The proposed approach is novel and general.

Weaknesses:
* Although the authors use 4 datasets and include languages more than English, they don't provide details of their specific domains and tasks, and looks all of the 4 datasets are for semantic parsing task in the spoken language domain. As the paper mentions, the sequence labeling tasks range from NER to QA. Therefore, the experiments are too narrow to support their claims generally on top of sequence labeling tasks.
* As a paper on sequence labeling tasks, the authors don't include the basic solution, token classification, as one of the baselines. Although this paper focuses on the generative manner, the performance of token classification should be still considered as important reference.
* Moreover, the token classification can also be performed using the same sequence-to-sequence backbone model, just like how BART (https://arxiv.org/pdf/1910.13461.pdf) does in the QA task (SQuAD), which should be also considered as an important baseline.

**Summary Of The Paper:**

This paper proposes GROOT,  a novel and general reward optimization framework for training sequence labeling models toward reward metrics, via construct corrective candidates as contrastive examples to guide the model learning.

**Summary Of The Review:**

Overall, the proposed method is novel and general, and paper is well-written, but the experiments cannot support the general claim because they lack task and domain diversity, and important baselines are missing.

---

> ### Author Response · Authors · 2022-11-20
> **Response to the comments**
>
> Thank you for taking the time to read our paper and sharing your insightful feedback. You have our sincere thanks and gratitude.
>
> Regarding your questions and comments, please see below:
>
>
> **Datasets and tasks**:
>
> Thank you for pointing this out. We have added descriptions of the datasets in Section 5.1 and to the corresponding table. As pointed out, the original four datasets are designed for semantic parsing / slot-filling but cover different domains from Travel, Dining, and Virtual Assistants.
>
> However your point was well-taken and we added results for two new datasets and tasks: NER and syntactic chunking in Appendix F.
>
> We hope that with the additional domains and tasks (now covering 8 different test sets), this may assuage any concerns you had for the breadth of applicability of our method.
>
> (We observed similar findings on a couple of new, challenging segmentation and parsing tasks; however, as these datasets are not publicly available yet, we refrained from reporting these results. Similarly we were hesitant to introduce results for a QA dataset as it would require explaining appropriate reward and correction functions. However, if you have suggestions for additional datasets / tasks please let us know.)
>
> **Token classification**:
>
> This is a fair point. We have added results for 3 additional baselines – including BERT and T5-based token classification methods – in Appendix H.
>
> A brief explanation of why we originally left these out:
> Our original experiments with GROOT were on a T5-based token-classification approach. However a recent paper (https://arxiv.org/abs/2203.08378) demonstrated that token-classification based approaches were significantly outperformed by the generative process. We thus redid our experiments based on their recommended setup. However, your question is a valid one which many readers may have, and hence we hope the new results put to rest any concerns regarding this comparison.
>
> We hope the above changes (coupled with the other experimental results and clarifications we’ve added) will help address the main concerns you had about our work. Please let us know if you have any further questions.

---

### Official Review · Reviewer_cdc3 · 2022-11-05

**Confidence:** 3
**Correctness:** 4
**Technical Novelty And Significance:** 3
**Empirical Novelty And Significance:** 3
**Recommendation:** 5

**Clarity, Quality, Novelty And Reproducibility:**

- The clarification can be clearer.
- The proposed method could help to improve the performance of the sequence labeling task.
- This work should be original.
- The paper presents experimental details for reproducibility.

**Strength And Weaknesses:**

### Pros
1. The proposed GROOT is simple, effective, and easy to implement.
2. The whole framework is clear.
3. The experimental results show great improvement in reward function and precision.

### Cons
1. The relationship and difference with *Shu++* should be discussed more clearly.
2. Only one machine translation baseline *Shu++* was compared. It will be more convincing if it is compared with more baselines.
3. Sufficient experiments and figures are provided, but there are no consistent performance gains across different datasets.
4. There should be more descriptions of the datasets.
4. Some statements are difficult to read and may contain some grammatical errors, e.g, *One may assume that just the above may be sufﬁcient for models to be learn the output space sufﬁciently as done in prior work (Shu et al., 2021)* on page 4, and *Correct erroneous candidate* paragraph on page 4, and *Works by replace incorrect tags in y with the correct ground-truth tags* on page 6. The authors should pay more attention to the writing.

**Summary Of The Paper:**

This paper proposed a new framework for training sequence labeling models to optimize reward metrics, and a CML loss to help the model better understand the reward space. The experimental results show superiority compared with recent baselines.

**Summary Of The Review:**

I'd like to assign 5 (marginally below the acceptance threshold). I hope the authors could make the clarification clearer and do more analysis on the influences of different modules/hyperparameters.

---

> ### Author Response · Authors · 2022-11-20
> **Response to the comments**
>
> Thank you for taking the time to read our paper and sharing your insightful feedback. You have our sincere thanks and gratitude.
>
> Regarding your questions and comments, please see below:
>
> **Relationship and difference with Shu++**:
>
> Thank you for bringing this up: This is a fair concern. While we previously had some discussions spread across the paper, we have expanded upon this particularly in Sections 2.2, Section 5.2 and Section 6 to clearly describe the relationship and difference.
>
> **Baselines other than Shu++**:
>
> As suggested by reviewer PagF, we have added 3 new baselines in Appendix H to provide additional insights into the benefits of the proposed approach.
>
> We would like to take a minute though to discuss why Shu++ is still the most important method to be compared against our proposed method. Shu et al. (2021) comprehensively discussed advantages of their proposed method against related methods (e.g., Shen et al. (2016) and Edunov et al. (2018)), and thus we believe that Shu++ is the best (baseline) representative among the reward optimization methods.
>
> **No consistent performance gains across different datasets**:
>
> Unfortunately we didn't quite understand your concern here. In general, we found that the improvement provided by the proposed method and the trends observed on varying different settings were all quite consistent across all six datasets.
>
> We should also stress here that our experiments were all conducted in a human-blind manner; that is, we did not refine any hyper-parameters or configurations using the test sets – which were only used once to report the final scores.
>
> While there are a couple of improvements which are not statistically significant, the overwhelming majority of numbers across the 8 test sets are significant – which we believe demonstrate a clear and consistent trend. For reference, Appendix F reports additional results on NER and chunking tasks, where we see the consistent effects of our method as well.
>
> **Dataset descriptions**:
>
> Thank you for pointing out this. We have added descriptions of the datasets in Section 5.1 and to Table 2.
>
> **General writing**:
>
> Thank you for pointing this out. While we tried our best to be as clear as possible (and even obtained feedback from colleagues to iron out the writing before submitting), we fully recognize that certain sections may have slipped through the cracks.
>
> We have taken another hard pass and aimed to polish the writing further within our allocated time (moving section 6.5 to the appendix so we have additional space for improving clarity).
>
> **More analysis on the influences of different modules/hyperparameters**:
>
> We were somewhat confused by this comment here as well. Sections 6.1-6.4 and Appendix E (+ Figures 2 to 11) together provide detailed analysis of virtually every key hyperparameter / model choice.
>
> For reference, as suggested by reviewer oMZW, Appendix G reports how robust our proposed framework is with different reward functions. We would like to know if there was a specific aspect you would like to see ablated or more deeply studied.
>
>
> We hope the above changes (coupled with the other experimental results and clarifications we’ve added) will help address the main concerns you had about our work. Please let us know if you have any further questions.

---

### Author Response · Authors · 2022-11-20
**Summary of updates in our paper**

We would firstly like to thank all the reviewers for their hard work and insightful feedback. Reviewing can at times feel like a thankless job and hence we wanted to extend our sincere thanks.  Your feedback has been invaluable as it has (hopefully) helped us produce a better paper.

While we address all your comments and questions in individual responses, we wanted to briefly summarize some of the major changes we made in this revision:

1. To demonstrate the wide applicability of our method, we have added results for additional datasets and tasks  (NER and chunking). As in the other experiments, all results were processed in a human-blind fashion; that is, we did not tune any hyperparameters or cherry-pick results to show robustness of our method.

2. We have added results for three additional baselines – including BERT and T5-based token classification approaches to the appendix.

3. To demonstrate that the method and findings are not specific to our original reward function, we added results for two additional reward functions.


Our previously stated empirical findings hold (in some cases with even larger gains) in these new experiments – illustrating the breadth and applicability of the method. Additionally from the exposition standpoint:

4. We have added details about the datasets and their domains / tasks they cover in both Table 2 and Section 5.1.

5. We better described the difference between Shu++ and our proposed method, and also provided additional intuition on why the proposed technique performs better.

6. Due to the page limit, we unfortunately had to move Section 6.5 to Appendix E.1 to satisfy the page limitation.

---

### Decision · Program_Chairs · 2023-01-20

**Decision:**

Reject

**Justification For Why Not Higher Score:**

Reviewers had concerns on the empirical studies, including lack of comparison with relevant baselines, lack of diversity of the evaluation datasets, missing details of the experiments, etc.

**Justification For Why Not Lower Score:**

The work is well motivated and the method is interesting and reasonably sound.

**Metareview: Summary, Strengths And Weaknesses:**

The paper proposes GROOT, a new framework for training sequence labeling models to optimize reward metrics. The approach constructs corrective candidates as contrastive examples and uses a margin loss. Experiments show improved performance on several datasets in different domains. Though the work is well motivated and the method is interesting, the reviewers had concerns on the empirical studies, including lack of comparison with relevant baselines, lack of diversity of the evaluation datasets, missing details of the experiments, etc.